# Multispectral Thermometry Method Based on Optimisation Ideas

**DOI:** 10.3390/s24072025

**Published:** 2024-03-22

**Authors:** Xuan Zhang, Bin Liu, Hongru Wang, Wen Ma, Yan Han

**Affiliations:** 1College of Information and Communication Engineering, North University of China, Taiyuan 030051, China; zhangxuanjohn@163.com (X.Z.); liubin414605032@163.com (B.L.); 2Shanxi Key Laboratory of Signal Capturing & Processing, North University of China, Taiyuan 030051, China; 3State Key Laboratory of Dynamic Testing Technology, North University of China, Taiyuan 030051, China; 4Shanxi TZ Digitization & Intelligence Technology Company, Taiyuan 030031, China; wanghongru@tz.com.cn (H.W.); mawen516@163.com (W.M.)

**Keywords:** multispectral, temperature measurement, optimisation, thermometry, blackbody radiation, temperature difference correlation function

## Abstract

Multispectral thermometry is based on the law of blackbody radiation and is widely used in engineering practice today. Temperature values can be inferred from radiation intensity and multiple sets of wavelengths. Multispectral thermometry eliminates the requirements for single-spectral and spectral similarity, which are associated with two-colour thermometry. In the process of multispectral temperature inversion, the solution of spectral emissivity and multispectral data processing can be seen as the keys to accurate thermometry. At present, spectral emissivity is most commonly estimated using assumption models. When an assumption model closely matches an actual situation, the inversion of the temperature and the accuracy of spectral emissivity are both very high; however, when the two are not closely matched, the inversion result is very different from the actual situation. Assumption models of spectral emissivity exhibit drawbacks when used for thermometry of a complex material, or any material whose properties dynamically change during a combustion process. To address the above problems, in the present study, we developed a multispectral thermometry method based on optimisation ideas. This method involves analysing connections between measured temperatures of each channel in a multispectral temperature inversion process; it also makes use of correlations between multispectral signals at different temperatures. In short, we established a multivariate temperature difference correlation function based on the principles of multispectral radiometric thermometry, using information correlations between data for each channel in a temperature inversion process. We then established a high-precision thermometry model by optimising the correlation function and correcting any measurement errors. This method simplifies the modelling process so that it becomes an optimisation problem of the temperature difference function. This also removes the need to assume the relationships between spectral emissivity and other physical quantities, simplifying the process of multispectral thermometry. Finally, this involves correction of the spectral data so that any impact of measurement error on the thermometry is reduced. In order to verify the feasibility and reliability of the method, a simple eight-channel multispectral thermometry device was used for experimental validation, in which the temperature emitted from a blackbody furnace was identified as the standard value. In addition, spectral data from the 468–603 nm band were calibrated within a temperature range of 1923.15–2273.15 K, resulting in multispectral thermometry based on optimisation principles with an error rate of around 0.3% and a temperature calculation time of less than 3 s. The achieved level of inversion accuracy was better than that obtained using either a secondary measurement method (SMM) or a neural network method, and the calculation speed achieved was considerably faster than that obtained using the SMM method.

## 1. Introduction

As a thermometry technique, non-contact radiation thermometry has been widely adopted in recent years because it does not require contact with the measured object [1,2,3,4,5,6,7]. In order to measure the transient surface temperature of combustion devices such as internal combustion engines and gas turbines, Sneha Neupane et al. developed a multi-infrared-channel pyrometry-based optical instrument for high-speed surface thermometry. Surface temperatures were obtained via non-linear least squares (NLLS) optimisation, based on multispectral radiation thermometry principles, using data for surface thermal radiation in four discrete spectral regions and a corresponding emissivity model. Experimental results showed that the instrument demonstrated excellent accuracy of >97% and 2-sigma precision of >99% in the 400–800 °C range, with a transient response of 20 µs in a bench validation test [8]. To analyse the damage to a target caused by a high-energy laser beam, Zhentao Wang et al. developed a multispectral thermal imager to measure the temperature field of the laser-induced damage. They used a five-channel, four-wavelength filter photometer, in which the fifth channel was used as a calibration channel, so that the infrared thermal imager could accurately collect data under multispectral channels. The spectral data collected by the instrument were inverted to determine the corresponding temperature field using the multispectral thermometry method, and the results showed that the measurement error of the multispectral thermal imager was <1.5% [9]. Ketui Daniel et al. implemented a multispectral method using a curve-fitting technique through simulation to measure the surface temperatures of ceramic coatings in the presence of high ambient radiation and low emissivity. Ten spectral emissivity models were selected in a certain wavelength range, and the simulated spectral data were fitted to each of the ten spectral emissivity models using the NLLS curve-fitting technique; models that produced poor fits of spectral radiance and spectral emissivity were then rejected, and predicted temperatures and emissivity coefficients were obtained. The experiments showed that, for zirconia with a coating thickness of 330 μm, the temperature prediction error was less than 1.5% when the ambient temperature was 1273 K and the target temperature was 800–1200 K [10]. To accurately measure the true surface temperature of an object in a high-temperature environment, Liwei Chen et al. proposed a multispectral thermometry method based on an adaptive emissivity model using a BP (back propagation) neural network to identify the shapes of spectral data; these shapes were then compared with those of commonly used emissivity models. The true temperature of the target was then determined after selection of the emissivity model that best conformed to the measured target and specific environment. Experimental results showed that, in the range of 575–685 °C, the maximum thermometry error using this method was 8 K after any influence of the high-temperature background was ignored [11]. In order to accurately measure the spectral emissivity, Wenjie Zhu et al. analysed and compared variations in spectral emissivity with wavelength for different temperatures and heating times by studying the analytical relationship between spectral emissivity and wavelength at different temperatures during the growth of an oxide layer on a specimen surface of 309S steel. Ten emissivity models were established, and these were used to achieve a true temperature inversion, with uncertainty within 10 K in the temperature range of 800–1100 K [12]. Xing et al. developed an emissivity range constraint optimisation data-processing algorithm in which emissivity had no effect. By fitting a large amount of data from different emissivity distribution target models, an effective search range for emissivity was obtained, greatly improving the efficacy of multispectral thermometry. Simulation and experimental results showed that the absolute error of temperature measurement was 25 K at a true temperature of 1800 K [13].

Although the above thermometry methods can predict radiant temperatures of targets, spectral emissivity remains unknown using all such techniques. Determination of spectral emissivity usually requires the assumption of particular mathematical relations for emissivity versus wavelength, or emissivity versus temperature, and any discrepancies between hypothesised and real-world situations seriously affect the inversion results. To address the above problems, Wang et al. proposed a novel thermometry method based on a constraint optimisation algorithm. First, they analysed the trend of spectral emissivity with wavelength increments. The trend in the emissivity–wavelength curve was then used to establish the constraints, and a genetic algorithm was used as an optimisation tool, without assuming a model of the spectral emissivity. Within a temperature range of 1200–3000 K, this method delivered high-precision thermometry with a relative error of less than 1% [14]; Xing et al. developed two new data-processing algorithms for multi-wavelength pyrometers: a gradient projection (GP) algorithm and an internal penalty (IPF) algorithm. The data-processing problem of the multi-wavelength pyrometer was transformed into a constraint optimisation problem, and either the GP or IPF algorithm was then used to estimate temperature, thus achieving inversion of the true target temperature without assuming spectral emissivity [15].

In summary, in the process of multispectral radiation thermometry, to achieve inversion of the true temperature, it is often necessary to assume the functional relationship between spectral emissivity and other physical quantities such as wavelength or true temperature. However, improved inversion results are achieved only when the spectral emissivity of the assumption model is close to the actual spectral emissivity of the object to be measured. Such a method is not applicable to the inversion of the true temperature of all materials. The constraint optimisation algorithm may be used to solve the problem described above, but this requires a narrow and fixed emissivity range as well as appropriate initial input values for emissivity; if these conditions are not satisfied, both accuracy and computational efficiency are greatly affected. Furthermore, when carrying out combustion thermometry of complex structural materials, such as special energy materials and fire explosives, the problem of complex materials arises. Differing ratios of component materials result in variations in radiation characteristics; this makes it difficult to use the spectral emissivity of the material in the static state to describe the characteristics of the radiation. In addition, in the process of combustion, the spectral emissivity of the material changes dynamically; even if it can be measured in a static state, this measurement still differs greatly from the actual value of spectral emissivity during combustion.

To better meet the demands of thermometry using complex new materials, for the present study, we developed a multispectral temperature measurement method based on multivariate extreme value optimisation, as described in [16]. This method avoids the process of assuming the relationship between spectral emissivity and other physical quantities without limiting the range of emissivity. By such means, the authors of [16] simplified the multispectral thermometry model. However, it did involve a certain level of measurement error in the process of true temperature inversion. To improve upon this previous work, then, we propose, in this paper, a multispectral thermometry method based on optimisation principles. Using the method described below, spectral data may be corrected without assuming the relationship between emissivity and other physical quantities, thus reducing the impact of measurement error. This results in more accurate inversion and faster calculation speed.

## 2. Principles of Temperature Inversion

In order to improve upon previous thermometry work, the error reported in [16] was analysed. It was determined that the errors in estimating the temperature and spectral emissivity could be determined using the expression $ln\left(\frac{\varepsilon_{\lambda_{i}}V_{\lambda_{i}b}}{V_{\lambda_{i}}}\right)$ and were mainly generated during the pyrometer calibration process, as shown in Figure 1. We therefore analysed this process as follows.

As can be seen in Figure 1, the pyrometer radiates the energy of the measured target at different wavelengths to each pixel of the photodetector and then converts the spectral energy into a voltage signal through photoelectric conversion. However, the spectra generated by the pyrometer are not completely linear. As a result, when calibrating the pyrometer, it is difficult to ensure the consistency of the data between various channels. In addition, distortion of the spectral line shape due to scattering phenomena causes crosstalk of energy between different detection channels, so the true voltage signal should be a value within the range of $\pm \Delta V_{\lambda }$. The same problem also affects wavelength calibration during the assembly of the photodetector, so the true wavelength should similarly have a value within the range of $\pm {\Delta \lambda }_{i}$.

According to [16], the measured temperature $T_{\lambda_{i}}$ indicated by the multi-wavelength pyrometer in channel i may be obtained as follows:(1)$$T_{\lambda_{i}}=1/\left(\frac{1}{T_{\lambda_{b}}}+\frac{\lambda_{i}}{C_{2}}ln\left(\frac{\varepsilon_{\lambda_{i}}V_{\lambda_{i}b}}{V_{\lambda_{i}}}\right)\right),$$

By analysing Equation (1) in conjunction with Figure 1, it can be seen that, in the process of temperature inversion, the values of $T_{\lambda_{b}},\;\lambda_{i},\;V_{\lambda_{i}b}$ and $V_{\lambda_{i}}$ have some measurement errors; the correction terms $\eta_{1}\Delta T_{\lambda_{b}}$**,**
$\eta_{2}\Delta \lambda_{i}$, $\eta_{3}\Delta V_{\lambda_{i}b}$ and $\eta_{4}\Delta V_{\lambda_{i}}$ are, therefore, introduced into $T_{\lambda_{b}}$, $\lambda_{i}$, $V_{\lambda_{i}b}$ and $V_{\lambda_{i}}$, respectively. We then obtain
(2)$$T_{\lambda_{i}}=1/\left(\frac{1}{T_{\lambda_{b}}+\eta_{1}\Delta T_{\lambda_{b}}}+\frac{\lambda_{i}+\eta_{2}\Delta \lambda_{i}}{C_{2}}ln\left(\frac{\varepsilon_{\lambda_{i}}\left(V_{\lambda_{i}b}+\eta_{3}\Delta V_{\lambda_{i}b}\right)}{V_{\lambda_{i}}+\eta_{4}\Delta V_{\lambda_{i}}}\right)\right),$$
where $\eta_{1}$, $\eta_{2}$, $\eta_{3}$ and $\eta_{4}$ are correction factors within the value range [–1,1] and $\Delta T_{\lambda_{b}}$, $\Delta \lambda_{i}$, $\Delta V_{\lambda_{i}b}$, $\Delta V_{\lambda_{i}}$ are the maximum measurement error. The corrected spectral data are shown in Figure 2. From the figure, it can be seen that the original data have coarse errors in many places, while the spectral curves after correction become smoother and closer to the real situation, further illustrating the importance of data correction.

### 2.1. Temperature Difference Function

The temperature of a given point on an object at a given time is theoretically unique. Consequently, as expressed in Equation (2), the temperatures $T_{\lambda_{i}}$ measured by different channels are equal, and the sum of the squares of the differences in the temperatures measured by different channels may be expressed as follows:(3)$${\left(T_{\lambda_{1}}-T_{\lambda_{3}}\right)}^{2}+{\left(T_{\lambda_{2}}-T_{\lambda_{4}}\right)}^{2}+\cdot \cdot \cdot +{\left(T_{\lambda_{n-2}}-T_{\lambda_{n}}\right)}^{2}+{\left(T_{\lambda_{n-1}}-T_{\lambda_{1}}\right)}^{2}=0,$$

Due to errors that are unavoidable during the measurement process, Equation (3) often does not hold true, so the temperature difference function consisting of the unknown $\varepsilon_{\lambda_{i}}$ is now introduced, as follows:(4)$${F=\left(T_{\lambda_{1}}-T_{\lambda_{3}}\right)}^{2}+{\left(T_{\lambda_{2}}-T_{\lambda_{4}}\right)}^{2}+\cdot \cdot \cdot +{\left(T_{\lambda_{n-2}}-T_{\lambda_{n}}\right)}^{2}+{\left(T_{\lambda_{n-1}}-T_{\lambda_{1}}\right)}^{2},$$
where the unknown $\varepsilon_{\lambda_{i}}$ is contained in $T_{\lambda_{i}}\left(i=1,2,\cdot \cdot \cdot ,n\right)$, as expressed in Equation (2). From relevant knowledge of the error, it can be seen that the smaller the value of the temperature difference function *F* (i.e., the smaller the difference between the temperatures measured by each channel), the closer the temperature $T_{\lambda_{i}}$ to the true temperature, and, thus, the greater the accuracy of the thermometry. In addition, the true temperature indicated by $T_{\lambda_{i}}$ is unique when *F* takes the minimum value of 0. However, due to the existence of errors, it is difficult for *F* to take the minimum value; moreover, in theory, there are infinitely many minimum values. The multispectral temperature inversion problem is, therefore, transformed into a function extreme value optimisation problem.

In the present study, there were n − 1 square terms in the temperature difference function *F*; in contrast, in the work described in [15], there were n square terms. We may say that function *F* was set more skillfully in the present work because it not only utilised the correlation information between different channels, but also reduced the amount of modelling data, thereby increasing the efficiency of the algorithm and improving upon the previous work.

### 2.2. Function Extreme Value Optimisation

From the relevant theory of spectral thermometry, it is known that the range of spectral emissivity in an object to be measured is (0,1), and that this relationship may be expressed as in Equation (5), as follows:(5)$$0<\varepsilon_{\lambda_{i}}<1,$$

This constraint relationship, although simple, constrains the range of values of $T_{\lambda_{i}}$ in Equation (2). Using this relationship, which constitutes an inequality constraint, a solution to the function extreme value optimisation problem may be obtained more quickly.

To solve the optimisation problem of the function under the inequality constraints, the relationship between the function *F* and the unknowns can be obtained by substituting Equation (2) into Equation (4); this may then be simplified for ease of representation, giving the following basic structural form:(6)$$\left\{\begin{array}{c} minF=\sum \left(\frac{1}{AX+B}-\frac{1}{CX+D}\right)\\ s.t.\;a\leq \varphi \left(X\right)\leq b \end{array}\right.,$$
where X denotes the n-dimensional vector consisting of the unknowns $\varepsilon_{\lambda_{i}}$; *A*, *B*, *C* and *D* are matrices of 1 × n, 1 × 1, 1 × n and 1 × 1, respectively; *minF* denotes the optimisation of the extreme value of the temperature difference function; and $a\leq \varphi \left(X\right)\leq b$ denotes the range of spectral emissivities. In the true temperature inversion process, a = 0 and b = 1. The optimisation code of the extreme value function can be obtained with the help of Particle Swarm, Gradient Descent, neural network and other algorithms. After the function is optimised to determine the extreme value, the spectral emissivity value can be obtained by calculation; this can then be re-substituted into the temperature difference function for the second optimisation. By such means, the inversion of the spectral emissivity, and the true temperature, may ultimately be obtained.

In order to clearly represent the thermometry work used in the present study, the function optimisation process may now be expressed in a step-by-step manner, as follows:
Modelling: In line with the principles of multispectral thermometry, as well as the principle of error correction, a temperature model is established, as expressed in Equation (2). The model not only avoids the error problem caused by inaccurate calibration of the system, but also corrects any measurement errors arising from the experimental process.Parameter determination: The following parameters need to be determined before function optimisation can be carried out:○Measurement error determination: $\Delta T_{\lambda_{b}}$ is generated in the process of temperature calibration, so the temperature error of the blackbody furnace, the temperature calibration device, is taken as the temperature measurement error. In addition, $\Delta \lambda_{i}$, $\Delta V_{\lambda_{i}b}$ and $\Delta V_{\lambda_{i}}$ are generated during assembly and usage of the spectral temperature measurement device, so it is necessary to determine the wavelength and voltage errors in combination with the spectral distribution of the temperature measurement device.○Determination of correction factors: In the process of assembling the thermometer, the coupling between the lens and the workpiece requires the use of optical glue and metal screws. This means that the relative positions of the spectrum and the detector, and the energy intensity between them, are not distributed as designed by the simulation. Measurement errors are also introduced by the scattering phenomenon and the working environment of the test. Factors causing measurement errors, such as the amounts of optical glue and metal material, were also recorded several times in the course of the present study. The amounts of optical glue and metal material, and all the resulting error data, were recorded and input into the neural network to establish a learning model and predict the values of the correction factors $\eta_{1}$, $\eta_{2}$, $\eta_{3}$ and $\eta_{4}$ in the range [−1,1].○Determination of initial emissivity value: From the relevant theory of spectral thermometry, it can be seen that the spectral emissivity of the object to be measured is in a range of (0, 1). In order to invert the target temperature without limiting the range of emissivity, and to prevent the accuracy of the thermometry from being dependent on the initial value of the emissivity, a randomly selected floating point number is used as the initial value of the emissivity in the range of (0, 1).

Model parameter selection is completed after measurement errors, correction factors and the initial emissivity value are determined.Function optimisation: Through the derivation of equations from Equations (3) and (4), the multispectral thermometry problem can now be transformed into a function extreme value optimisation problem. The optimisation function and constraints are shown in Equation (6). The function extreme value optimisation code can be implemented with the help of algorithms such as Particle Swarm, Gradient Descent and neural network.Secondary optimisation: After function optimisation is used to determine the extreme value, the spectral emissivity value can be obtained by calculation; this can be re-substituted into the temperature difference function for the second optimisation. By such means, the inversion of the spectral emissivity and the true temperature may ultimately be obtained.

With the establishment of the model, the determination of parameters, the function optimisation and the secondary optimisation, the temperature measurement work of the present study was completed.

## 3. Experimental Validation

### 3.1. Experimental Setup

To verify the thermometry method proposed in this paper, the application of transient combustion temperature radiation measurement of fire explosive is taken as an example. To this end, we adopted the multispectral thermometry device described in [17], as illustrated in Figure 3a. The spectrometer decomposes the radiation generated by the blackbody furnace into spectral segments of different wavelengths, then converts them into electrical signals. The multispectral data are then converted and transmitted to a PC for temperature estimation. Based on the high temperature distribution range of the fire explosive during transient combustion, the temperature of the blackbody furnace was set to 1923.15 K, 1973.15 K, 2023.15 K, 2073.15 K, 2123.15 K, 2173.15 K, 2223.15 K and 2273.15 K, successively. After the temperature was stabilised, the spectrometer was used to collect the spectral data at a distance of 0.5 m from the blackbody furnace window. Calculations based on the spectral distribution reported in [17] showed that the measurement errors $V_{\lambda }$ and $\lambda_{i}$ were 4.88% and 2.6%, respectively. The temperature generated by the blackbody furnace was determined to be the standard value in the experiment, and spectral data were calibrated and corrected on this basis. After spectral emissivity was determined with the help of algorithms such as Particle Swarm, Gradient Descent and neural network, the spectral emissivity was re-substituted into the multivariate temperature function for the secondary optimisation. Finally, the inversion of the spectral emissivity and the true temperature was determined. The process is shown in Figure 3b.

The blackbody furnace model LS3000-100, produced by Electro Optical, was used as the standard radiation source to calibrate the system. It has a radiation temperature range of 1000–3000 °C, emissivity of 0.997 ± 0.002, a spectral radiation range of 0.2–11 µm and a temperature error of 0.25% up to 2700 °C, i.e., the value of $\Delta T_{\lambda_{b}}$ is 4.125 °C at 1650 °C (1923.15 K). The spectral data were processed on a PC produced by Lenovo, model Thinkpad E540, with an Intel(R)Core(TM) i5-4210M processor (CPU 2.60 GHz) and 8.00 GB of installed memory.

The effective wavelengths of the 8 channels of the spectrometer and the voltage data for all channels at the reference temperature of 1923.15 K are shown in Table 1. The voltage data for all channels at different measurement temperatures are shown in Table 2.

### 3.2. Results

A temperature inversion method based on optimisation principles was used to determine spectral emissivity and inversion temperature; the calculation time was then recorded. The average value of the measured temperatures of eight channels was then taken as the measurement result, as shown in Table 3. It can be seen that, in the temperature range of 1923.15–2273.15 K, the measurement error is basically less than 0.5%, with an error range of 0.04–0.57% and calculation times of less than 3 s.

The effectiveness of the method used in the present study can now be compared with the secondary measurement method (SMM method) described in [18]. From the comparison results set out in Table 3, it can be seen that our method delivered greater accuracy than the SMM method, with the measurement error range of the SMM method being between 0.05% and 2.07%. In addition, the SMM method required a calculation time of more than 55 s; using the method described in this paper, calculation time was reduced to less than 3 s under the same conditions. Moreover, the calculation speed was greatly improved.

From Table 3, it can be seen that, using the SMM method, initial temperatures were assigned values which were 20 K higher than the first temperature point. Measurements at 1943.15 K, 2043.15 K, 2143.15 K and 2243.15 K were then taken, giving eight temperatures in total. The measurement data for the SMM method and the method proposed in this paper were both taken from the spectral data of blackbody furnace radiation, and both methods were run on the same PC, so that the experimental conditions were the same.

The method used in the present study was also compared with a thermometry method based on a neural network, and the results are shown in Table 4. From the comparison results in Table 4, it can be seen that, in the temperature range of 1923.15–2273.15 K, the measurement errors obtained using the method proposed in this paper are lower than those using the neural network-based method. We may say, therefore, that our method delivered thermometry that was more accurate than the neural-network-based thermometry method in the temperature range of 1923.15K–2273.15 K. However, no comparison could be obtained with respect to thermometry time because the neural network requires a large amount of data for training, resulting in time-consuming data processing, thereby rendering any comparison between the two methods meaningless.

### 3.3. Advantages of the Method Proposed in this Paper

From the above experimental results, it can be seen that the method proposed in this paper avoids the process of assuming the relationship between spectral emissivity and other physical quantities; it also simplifies the multispectral thermometry model and exhibits the advantages of high inversion accuracy and fast calculation speed. In addition to the above features, and in comparison with previously reported methods [9,11], the method newly described here does not require limiting the emissivity range; this may be seen as an advantage, as it greatly improves the possibility of exploring the radiative properties of unknown materials or new composites. Moreover, the methods described in [9,11] both require large amounts of data to simulate the emissivity range; in this respect, the method used in the present study may be seen as more convenient. Finally, the method described in this paper offers an obvious advantage in terms of calculation speed compared with the method described in [11].

The feasibility of the method proposed in this paper is, therefore, verified by the experimental results described above; in short, the method reported here delivers improvements in terms of thermometry accuracy and calculation speed compared with other methods.

## 4. Discussion

### Effect of Spectral Data Correction on Spectral Emissivity Solution

Figure 4 shows curves for spectral emissivity versus wavelength obtained at different temperatures using the method described in this paper. In the figure, ‘‘Real value’’ represents the curve for real spectral emissivity versus wavelength. Figure 2 shows a spectral emissivity curve obtained by spectral data without correction. It can be seen that the emissivity value varies linearly with the increase in wavelength but is very different from the real value. Figure 2 also shows the curve after spectral data were corrected. This shows that, although there is an error in the estimation of spectral emissivity in the band of 450–500 nm, the estimation of spectral emissivity is significantly more accurate after the data correction. By analysing the spectral distribution reported in [17], we may conclude that the error in the 450–500 nm band was caused by the inconsistency in the data between different channels of the spectrometer, so the overall conclusion of this paper is not affected.

In order to investigate the effects of the measurement errors $\Delta T_{\lambda_{b}}$, $\Delta \lambda_{i}$, and $\Delta V_{\lambda }$ on the determination of spectral emissivity, the following tests were carried out during the experimental process: (1) estimation of spectral emissivity after correcting $\lambda_{i}$ and $V_{\lambda }$; (2) estimation of spectral emissivity after correcting $T_{\lambda_{b}}$ and $V_{\lambda }$; (3) estimation of spectral emissivity after correcting $T_{\lambda_{b}}$ and $\lambda_{i}$. The results of these tests are shown in Figure 5. 

Analysis of Figure 5a reveals estimations of spectral emissivity, which are close to the real values overall; however, the uncorrected $T_{\lambda_{b}}$ results in accuracy results which are slightly lower than the results shown in Figure 4b. This indicates that correcting $T_{\lambda_{b}}$ improves accuracy in the estimation of spectral emissivity, and further indicates that there is a certain amount of error in the measurement process of $T_{\lambda_{b}}$. In terms of experimental results, Figure 5b may be seen as the same as Figure 4b, with a measurement error of $\lambda_{i}$ which is very small, indicating that adoption of the optical structure described in [17] results in accurate calibration of wavelength. However, the result at 468 nm is slightly worse in Figure 4b than in Figure 5b after correcting $\lambda_{i}$; this indicates that correcting $\lambda_{i}$ introduces a certain amount of error, and further confirms the precision of the optical structure described in [17]. It can also be seen that the effect of an uncorrected $V_{\lambda }$ on the accuracy of experimental estimation is very large, indicating a large error in the voltage measurement, further confirming the efficacy of the method used in the present study.

From the above discussion, it may be concluded that the method proposed in this paper can reduce the impact of experimental errors on results in the process of thermometry.

## 5. Conclusions

The multispectral thermometry method based on optimisation principles which is proposed in this paper avoids the errors associated with assumption models of spectral emissivity; it also simplifies a complex modelling process into an optimisation problem of multispectral temperature difference, improves modelling efficiency, corrects spectral data and reduces the impact of measurement errors on experimental results. We suggest, therefore, that it is more convenient for application in engineering. Experiments in the temperature range of 1923.15–2273.15 K showed that the method proposed in this paper delivered improved thermometry accuracy, compared with a secondary measurement method and a neural network method. The inversion accuracy had an error rate of about 0.3%, and calculation speed was substantially improved, compared with the secondary measurement method, with a temperature calculation time of less than 3 s.

## Figures and Tables

**Figure 1 sensors-24-02025-f001:**
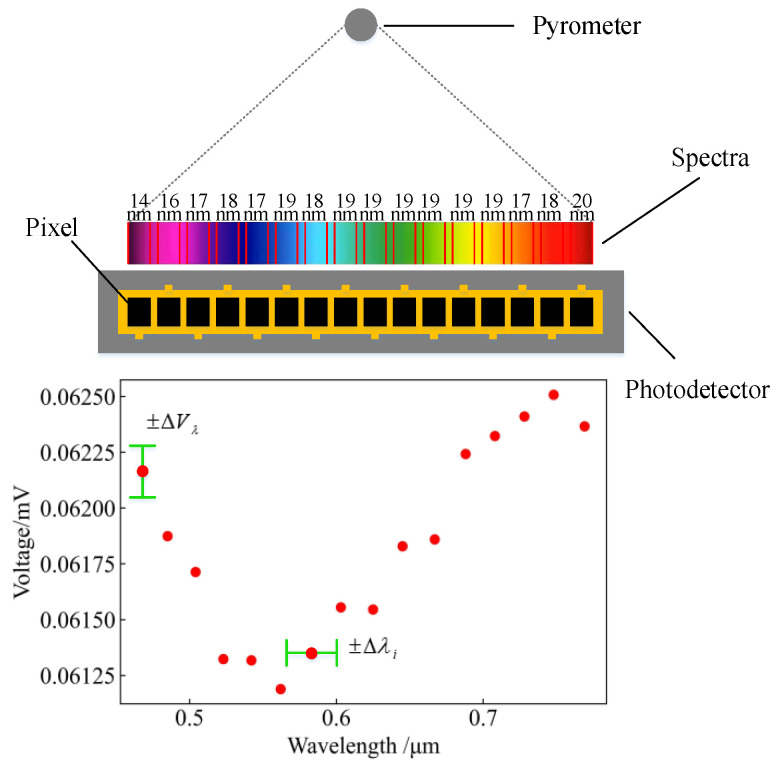
Pyrometer calibration process.

**Figure 2 sensors-24-02025-f002:**
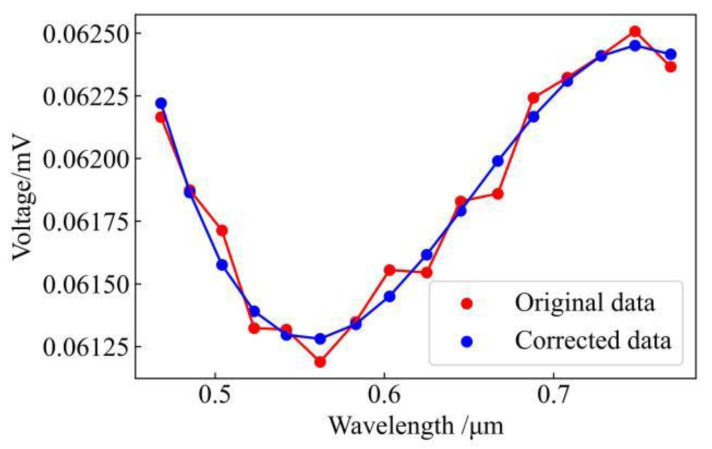
Comparison of spectral data before and after correction.

**Figure 3 sensors-24-02025-f003:**
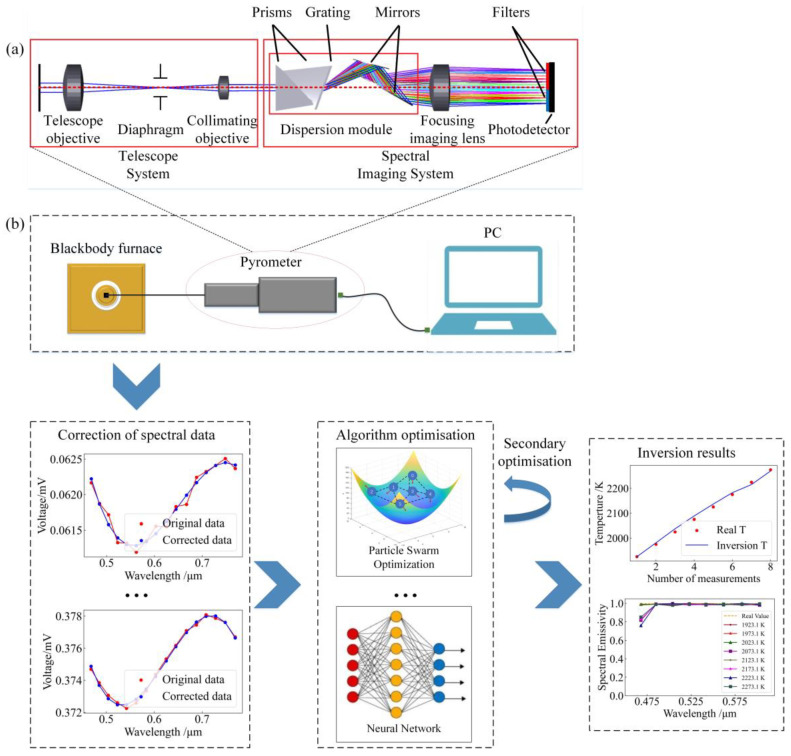
Experimental structure diagram: (**a**) pyrometer; (**b**) experimental procedure.

**Figure 4 sensors-24-02025-f004:**
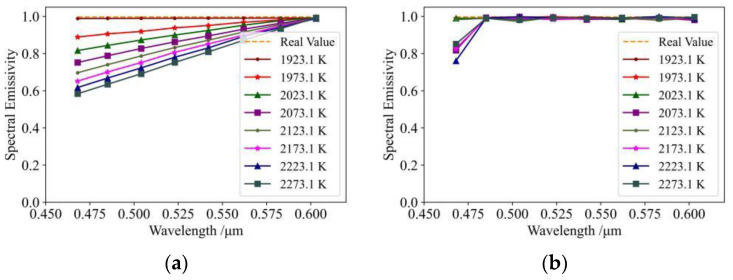
Variation trend of spectral emissivity with wavelength: (**a**) results of uncorrected spectral data; (**b**) results of corrected spectral data.

**Figure 5 sensors-24-02025-f005:**
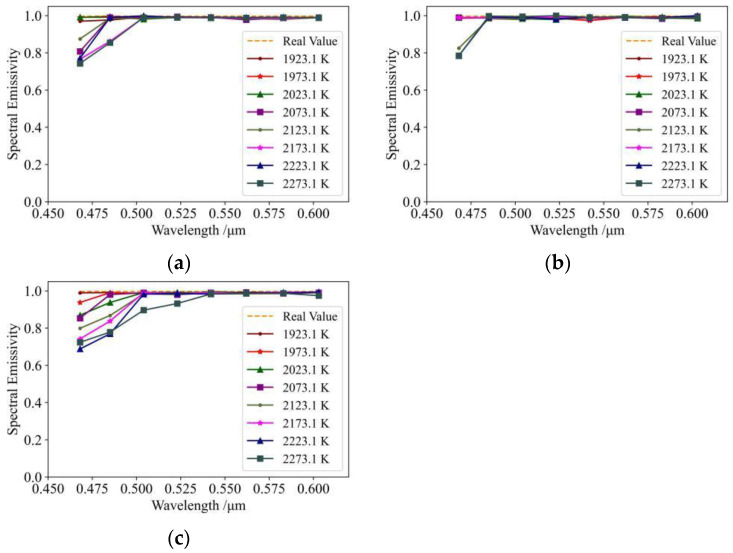
Effect of measurement error on spectral emissivity results: (**a**) spectral emissivity results after correction for $\lambda_{i}$ and $V_{\lambda }$; (**b**) spectral emissivity results after correction for $T_{\lambda_{b}}$ and $V_{\lambda }$; (**c**) spectral emissivity results after correction for $T_{\lambda_{b}}$ and $\lambda_{i}$.

**Table 1 sensors-24-02025-t001:** Voltage values for each channel at each reference temperature.

Channel	1	2	3	4	5	6	7	8
Effective wavelength/nm	468	485	504	523	542	562	583	603
Output voltage/V	0.0622	0.0619	0.0622	0.0613	0.0613	0.0612	0.0613	0.0616

**Table 2 sensors-24-02025-t002:** Voltage value of each channel under measurement temperature.

No.	Temperatures/K	CH 1	CH 2	CH 3	CH 4	CH 5	CH 6	CH 7	CH 8
1	1923.15	0.0622	0.0619	0.0622	0.0613	0.0613	0.0612	0.0613	0.0616
2	1973.15	0.0887	0.0885	0.0880	0.0879	0.0877	0.0879	0.0880	0.0881
3	2023.15	0.1205	0.1201	0.1197	0.1194	0.1192	0.1194	0.1196	0.1198
4	2073.15	0.1569	0.1566	0.1562	0.1556	0.1555	0.1556	0.1560	0.1562
5	2123.15	0.2022	0.2019	0.2014	0.2008	0.2006	0.2007	0.2012	0.2016
6	2173.15	0.2520	0.2515	0.2508	0.2506	0.2501	0.2503	0.2510	0.2514
7	2223.15	0.3087	0.3081	0.3072	0.3069	0.3063	0.3070	0.3074	0.3081
8	2273.15	0.3747	0.3738	0.3731	0.3726	0.3722	0.3726	0.3733	0.3744

**Table 3 sensors-24-02025-t003:** Comparison of inversion results obtained using the method proposed in this paper and the SMM method.

No.	Temperatures/K	Method1 Results/K	Method 1 Error	Method1 Time/s	Initial Value of SMM/K	SMM Results/K	SMM Error	SMM Time/s
1	1923.15	1923.88	0.04%	2.88	1943.15	1962.02	2.02%	56.56
2	1973.15	1982.61	0.48%	2.83		2013.88	2.07%	56.56
3	2023.15	2029.63	0.32%	2.62	2043.15	2055.91	1.62%	56.13
4	2073.15	2084.94	0.57%	2.43		2097.54	1.18%	56.13
5	2123.15	2133.85	0.50%	2.49	2143.15	2147.94	1.17%	55.19
6	2173.15	2179.36	0.29%	2.73		2185.46	0.57%	55.19
7	2223.15	2224.94	0.09%	2.42	2243.15	2238.61	0.70%	58.52
8	2273.15	2271.90	0.05%	2.47		2274.35	0.05%	58.52

Note. Method 1 denotes the method proposed in this paper.

**Table 4 sensors-24-02025-t004:** Comparison of inversion results obtained using the method proposed in this paper and the neural network method.

No.	Temperatures/K	Method 1 Results/K	Method1 Error	Method 1 Time/s	Method 2 Results/K	Method 2 Error
1	1923.15	1923.88	0.04%	2.88	1937.87	0.76%
2	1973.15	1982.61	0.48%	2.83	1994.33	1.06%
3	2023.15	2029.63	0.32%	2.62	2049.75	1.29%
4	2073.15	2084.94	0.57%	2.43	2120.43	2.22%
5	2123.15	2133.85	0.50%	2.49	2147.21	1.12%
6	2173.15	2179.36	0.29%	2.73	2192.97	0.90%
7	2223.15	2224.94	0.09%	2.42	2242.23	0.85%
8	2273.15	2271.90	0.05%	2.47	2294.99	0.95%

Note. Method 1 denotes the method in this paper; Method 2 denotes the neural-network-based thermometry method.

## Data Availability

The data presented in this study are available on request from the corresponding author.

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
