# Peer review of "Multispectral Thermometry Method Based on Optimisation Ideas"

_sensors, 2024, doi:10.3390/s24072025_

Round 1
Reviewer 1 Report
Comments and Suggestions for Authors
I was really exited to read an interesting manuscript about "Multi-spectral thermometry method based on optimisation ideas" as the title says, but I was more and more disappointed while reading it. The level of English grammar is quite low. Despite of the sentences do make sense in terms of syntax, they very often do not make sense in terms of comprehension. I would say, that I am quite familiar with the topic of non contact thermometry/thermovision, but I was very confused while reading the manuscript. I think that the most problematic part is the lack of information through the manuscript. Authors are writing a lots of text describing the results, but a lot of information is missing in description of calculation methodology. Maybe the key problem was that authors are continuing research described in reference 20, but I was not able to get that paper from the publisher, as literally everything on that page was written in Chinese and as I understood it required some authorization. I am missing basic explanation of some variables used in equations, for example V_lambda_i_b or C2 is a big mystery for me. Because of such misunderstandings, I was not able to follow the calculation methodology and therefore I am not able to judge the relevancy and significance of the method. Authors have introduced quite a number of correction factors which are calculated using several advanced optimization algorithms, but the conditions for the calculation were unclear for me. This is a job for math expert, not for expert in spectroscopy and therefore it should have been better explained for publication in Sensors journal. The calculation of the correction factors is a "black box" in the experimental results evaluation and I doubt that anyone outside the authors team would be able to repeat the calculations just using information mentioned the manuscript. The experimental part is also quite confusing. I really did not find any explanation of secondary measurement method, so I cannot trust the results. At the end the results seem to be very interesting, but I cannot approve the methodology used.
Therefore I cannot recommend the manuscript to be accepted before major changes to the content are made.
Comments on the Quality of English LanguageThe quality of English language is very poor. Sometimes the whole context is missing in the sentences. Authors do not use common scientific terms and it is than very hard to extract the real meaning of the sentences. Just as an example, they use the term "inversion" through all the manuscript, while it shall be obviously "conversion" of temperature from voltage. It needs a comprehensive proof check.
Author Response
Dear Editor and Reviewers:
Thank you for your letter and the reviewer’s comments concerning our manuscript. Those comments are valuable and very helpful for revising and improving our paper. We have studied the comments carefully and have made corrections which we hope meet with approval. Modified sections are marked in the red-lined version of the manuscript.
The line number involved below refer to the latest manuscript. For ease of reference, we have also updated line numbers in the comments. The specific modifications are as follows:
--------Responds to the reviewer’s comments--------
Reviewer 1:
Major comments:
- Comment: I was really exited to read an interesting manuscript about "Multi-spectral thermometry method based on optimisation ideas" as the title says, but I was more and more disappointed while reading it. The level of English grammar is quite low. Despite of the sentences do make sense in terms of syntax, they very often do not make sense in terms of comprehension. I would say, that I am quite familiar with the topic of non contact thermometry/thermovision, but I was very confused while reading the manuscript. I think that the most problematic part is the lack of information through the manuscript. Authors are writing a lots of text describing the results, but a lot of information is missing in description of calculation methodology. Maybe the key problem was that authors are continuing research described in reference 20, but I was not able to get that paper from the publisher, as literally everything on that page was written in Chinese and as I understood it required some authorization. I am missing basic explanation of some variables used in equations, for example V_lambda_i_b or C2 is a big mystery for me. Because of such misunderstandings, I was not able to follow the calculation methodology and therefore I am not able to judge the relevancy and significance of the method. Authors have introduced quite a number of correction factors which are calculated using several advanced optimization algorithms, but the conditions for the calculation were unclear for me. This is a job for math expert, not for expert in spectroscopy and therefore it should have been better explained for publication in Sensors journal. The calculation of the correction factors is a "black box" in the experimental results evaluation and I doubt that anyone outside the authors team would be able to repeat the calculations just using information mentioned the manuscript. The experimental part is also quite confusing. I really did not find any explanation of secondary measurement method, so I cannot trust the results. At the end the results seem to be very interesting, but I cannot approve the methodology used.
Therefore I cannot recommend the manuscript to be accepted before major changes to the content are made.
Response to ”The level of English grammar is quite low. ... . I think that the most problematic part is the lack of information through the manuscript.”: Thank you for pointing this out. We recognize that our manuscript should undergo extensive English revisions and have engaged a professional retouching agency to address this issue during revision.
Response to “I would say, that I am quite familiar with the topic of non contact thermometry/thermovision, ... , I was not able to follow the calculation methodology and therefore I am not able to judge the relevancy and significance of the method.”: Thanks for your careful review. In order to enable reviewers and readers to better understand the work of the manuscript, we have provided the original text of reference 16. Please see the attachment.
Response to “Authors have introduced quite a number of correction factors which are calculated using several advanced optimization algorithms, ... . The calculation of the correction factors is a "black box" in the experimental results evaluation and I doubt that anyone outside the authors team would be able to repeat the calculations just using information mentioned the manuscript. ”: We apologize for the lack of clarity. In the process of assembling the thermometer, the coupling between the lens and the workpiece requires the use of optical glue and metal screws, which makes the relative position and energy intensity between the spectrum and the detector are not distributed as designed by the simulation. In addition, scattering phenomenon and test working environment are also factors that introduce measurement errors. Factors causing measurement errors, such as the amount of optical glue and metal material, were recorded several times. The maximum voltage measurement error and wavelength calibration error in the calibration test were taken as , , , in Eq.(2), and the amount of optical glue, metal material and error data recorded each time were input into the neural network to establish a learning model and predict the value of the correction factor , , , . Therefore, the value of the correction factor depends on the result predicted by the neural network. We've restated this in line 246-259.
Response to “The experimental part is also quite confusing. I really did not find any explanation of secondary measurement method, so I cannot trust the results. At the end the results seem to be very interesting, but I cannot approve the methodology used.”: We apologize for the lack of clarity. In order for reviewers and readers to have a better understanding of the secondary measurement method, we have attached the original report of the secondary measurement method to the references 18: Sun X.G.; Yuan G.B.; Dai J.M.; Chu Z.X. Processing Method of Multi-Wavelength Pyrometer Data for Continuous Temperature Measurements. Instrumentation Science and Technology 2005, 26, 1255-1261.

Reviewer 2 Report
Comments and Suggestions for Authors
Title: Multi-spectral thermometry method based on optimisation ideas
Authors: Xuan Zhang, Bin Liu, Hong-Ru Wang, Wen Ma,Yan Han
Improving the accuracy of temperature measurements is a problem that has worried researchers and engineers for many years. Despite the progress of modern pyrometers, the accuracy of measurements has not yet reached the limit. In this regard, the paper by X. Zhang et al. is timely. Previously, the authors suggested the multi-spectral temperature measurement method based on multivariate extreme value optimization [Zhang X. et al., Spectroscopy and Spectral Analysis, 2023]. This paper describes the optimized multi-spectral thermometry method.
In my opinion, this manuscript can be recommended for publication with minor revisions.
There are some questions that need clarification:
1. In practice, the temperature measurement accuracy and precision may decrease due to scattering. How will distortion of the spectral line shape due to scattering phenomena affect the temperature difference function F?
2. Page 4, Eq.(2); pages 7, lines 221-223:
The values of the four correction factors were determined as numbers randomly generated in the range of [-1,1]. What do they really depend on?
3. Page 13, lines 376-377:
“...the voltage measurement has a large error, which can be reduced but unavoidable by optimising the photodetection circuit, which further illustrates the importance of the method in this paper”.
It is not entirely clear how the proposed method is related to the obvious need to extremely optimize the PD circuit before any measurements.
4. Page 1, line 22:
“ideasa” - not English, but Scottish
Author Response
Dear Editor and Reviewers:
Thank you for your letter and the reviewer’s comments concerning our manuscript. Those comments are valuable and very helpful for revising and improving our paper. We have studied the comments carefully and have made corrections which we hope meet with approval. Modified sections are marked in the red-lined version of the manuscript.
The line number involved below refer to the latest manuscript. For ease of reference, we have also updated line numbers in the comments. The specific modifications are as follows:
--------Responds to the reviewer’s comments--------
Reviewer 2:
Major comments:
- Comment: In practice, thetemperature measurement accuracy and precision may decrease due to How will distortion of the spectral line shape due to scattering phenomena affect the temperature difference function F?
Response: Thank you for pointing this out. In practice, the temperature measurement accuracy and precision can indeed decrease due to scattering. Distortion of the spectral line shape due to scattering phenomena will cause crosstalk of energy between different detection channels, which results in voltage measurement errors and wavelength calibration errors as shown in Figure 1 in the manuscript, and further lead to inaccurate measured temperatures in each channel and increase F-value, which is exactly the problem that can be solved by the optimization function F in this manuscript. And we supplemented the description in line 160-161.
- Comment: Page 4, Eq.(2);pages 7, lines 221-223: The values of the four correction factors were determined as numbers randomly generated in the range of [-1,1]. What do they really depend on?
Response: We apologize for the lack of clarity. In the process of assembling the thermometer, the coupling between the lens and the workpiece requires the use of optical glue and metal screws, which makes the relative position and energy intensity between the spectrum and the detector are not distributed as designed by the simulation. In addition, scattering phenomenon and test working environment are also factors that introduce measurement errors. Factors causing measurement errors, such as the amount of optical glue and metal material, were recorded several times. The maximum voltage measurement error and wavelength calibration error in the calibration test were taken as , , , in Eq.(2), and the amount of optical glue, metal material and error data recorded each time were input into the neural network to establish a learning model and predict the value of the correction factor , , , . Therefore, the value of the correction factor depends on the result predicted by the neural network. We've restated this in line 229-238 and reformulated the equation(2) in line 171-172.
- Comment:Page 13,lines 376-377:“...the voltage measurement has a large error, which can be reduced but unavoidable by optimising the photodetection circuit, which further illustrates the importance of the method in this paper”.It is not entirely clear how the proposed method is related to the obvious need to extremely optimize the PD circuit before any measurements.
Response:Thank you for your significant reminder. The expression “...the voltage measurement has a large error, which can be reduced but unavoidable by optimising the photodetection circuit, which further illustrates the importance of the method in this paper” is amended to read “...the voltage measurement has a large error, which further illustrates the importance of the method in this paper” in line 421-424.
- Comment:Page 1, line 22:“ideasa” - not English, but Scottish
Response:Thank you for your significant reminder. We revised the original description in line 22.
---------------------------------------------------------------
In addition, we revised the format of some references. We have tried our best to improve the manuscript and made some changes to the manuscript. These changes will not influence the content and framework of the paper.
We earnestly appreciate for Editors/Reviewers’ warm work and hope that the correction will meet with approval. Thank you again for your positive and constructive comments and suggestions on our manuscript.
Sincerely,
Yan Han
Shanxi Key Laboratory of Signal Capturing and Processing, North University of China Taiyuan, Shanxi 030051, China
+8613603536145
Email: hanyan@nuc.edu.cn